# The p.R66W Variant in *RAC3* Causes Severe Fetopathy Through Variant-Specific Mechanisms

**DOI:** 10.3390/cells13232032

**Published:** 2024-12-09

**Authors:** Ryota Sugawara, Hidenori Ito, Hidenori Tabata, Hiroshi Ueda, Marcello Scala, Koh-ichi Nagata

**Affiliations:** 1Department of Molecular Neurobiology, Institute for Developmental Research, Aichi Developmental Disability Center, 713-8 Kamiya, Kasugai 480-0392, Japan; suga.ryo2018@gmail.com (R.S.); itohide@inst-hsc.jp (H.I.); tabata@inst-hsc.jp (H.T.); 2United Graduate School of Drug Discovery and Medical Information Sciences, Gifu University, Yanagido 1-1, Gifu 501-1193, Japan; ueda.hiroshi.j6@f.gifu-u.ac.jp; 3Center for One Medicine Innovative Translational Research (COMIT), Gifu University, 1-1 Yanagido, Gifu 501-1193, Japan; 4Department of Neurosciences, Rehabilitation, Ophthalmology, Genetics, Maternal and Child Health, University of Genoa, 16147 Genoa, Italy; marcelloscala87@gmail.com; 5Department of Neurochemistry, Nagoya University Graduate School of Medicine, 65 Tsurumai-cho, Nagoya 466-8550, Japan

**Keywords:** RAC3, small GTPase, neurodevelopmental disorder, corticogenesis, neuronal migration, axon extension

## Abstract

*RAC3* encodes a small GTPase of the Rho family that plays a critical role in actin cytoskeleton remodeling and intracellular signaling regulation. Pathogenic variants in *RAC3*, all of which reported thus far affect conserved residues within its functional domains, have been linked to neurodevelopmental disorders characterized by diverse phenotypic features, including structural brain anomalies and facial dysmorphism (NEDBAF). Recently, a novel de novo *RAC3* variant (NM_005052.3): c.196C>T, p.R66W was identified in a prenatal case with fetal akinesia deformation sequence (a spectrum of conditions that interfere with the fetus’s ability to move), and complex brain malformations featuring corpus callosum agenesis, diencephalosynapsis, kinked brainstem, and vermian hypoplasia. To investigate the mechanisms underlying the association between RAC3 deficiency and this unique, distinct clinical phenotype, we explored the pathophysiological significance of the p.R66W variant in brain development. Biochemical assays revealed a modest enhancement in intrinsic GDP/GTP exchange activity and an inhibitory effect on GTP hydrolysis. Transient expression studies in COS7 cells demonstrated that RAC3-R66W interacts with the downstream effectors PAK1, MLK2, and N-WASP but fails to activate SRF-, AP1-, and NFkB-mediated transcription. Additionally, overexpression of RAC3-R66W significantly impaired differentiation in primary cultured hippocampal neurons. Acute expression of RAC3-R66W in vivo by in utero electroporation resulted in impairments in cortical neuron migration and axonal elongation during corticogenesis. Collectively, these findings suggest that the p.R66W variant may function as an activated version in specific signaling pathways, leading to a distinctive and severe prenatal phenotype through variant-specific mechanisms.

## 1. Introduction

The Rac subfamily, a branch of the Rho family of small GTPases, includes RAC1, RAC2, and RAC3, which share approximately 90% amino acid sequence identity and engage common downstream effectors [1,2,3]. These GTPases are crucial regulators of actin cytoskeleton dynamics and intracellular signaling, influencing cell morphology, migration, adhesion, and the cell cycle in various cellular contexts [4,5]. RAC1 is ubiquitously expressed, while RAC2 is mainly found in hematopoietic cells, and RAC3 is predominantly expressed in the brain [6,7,8]. Notably, RAC3 is integral to several key aspects of neuronal development, including neurite extension, axonal and dendritic formation, synaptogenesis, and neuronal migration [9,10,11,12,13]. Like other small GTPases, RAC3 alternates between an active state bound to GTP and an inactive state bound to GDP. These transitions are governed by structural changes in the Switch I and Switch II regions, which are essential for effector recognition and GTP-dependent signaling. RAC3 activity is finely controlled by guanine nucleotide exchange factors (GEFs), which facilitate GDP/GTP exchange, activating RAC3, and GTPase-activating proteins (GAPs), which promote GTP hydrolysis, leading to RAC3 inactivation [14].

RAC1, RAC3, and CDC42 (another Rho family member) are critical regulators of brain development [4], and disruptions in their signaling pathways have been implicated in various neurodevelopmental disorders (NDDs) [15,16]. Besides the involvement of *RAC1*, *RAC3*, and *CDC42* variants in multiple NDDs [17,18,19,20,21,22,23,24], the Simons Foundation Autism Research Initiative (SFARI) database has identified several GAPs, GEFs, and downstream effectors as candidate genes for autism spectrum disorders (ASDs) [25]. To date, 12 de novo missense variants in *RAC3* have been associated with a clinically heterogeneous disorder known as NEurodevelopmental Disorder with Brain Anomalies and dysmorphic Facies (NEDBAF, MIM #618577) [19,20,21,22].

The novel de novo *RAC3* variant (NM_005052.3): c.196C>T, p.R66W was recently identified via whole exome sequencing (WES) in a male fetus at 24 weeks gestation, presenting with akinesia deformation sequence, which involves a range of conditions arising from genetic, neuromuscular, or other abnormalities that impair fetal movement. The fetus also exhibited complex brain malformations, including corpus callosum agenesis, diencephalosynapsis, kinked brainstem, and vermian hypoplasia [26]. This variant affects the Switch II region of RAC3 and causes a peculiar and particularly severe “fetopathy” which differs from typical NEDBAF phenotypes, raising the possibility that specific molecular disruptions may be involved in its pathogenesis.

In this study, we investigated the pathophysiological mechanisms underlying the association of the p.R66W variant with such severe prenatal phenotype. Biochemical and cell biological analyses revealed that the p.R66W variant functions as a constitutively active form in vitro, interacting with downstream effectors such as PAK1 (p21-activated kinase 1), MLK2 (mixed-lineage kinase 2), and N-WASP (neuronal Wiskott–Aldrich syndrome protein). However, this variant showed minimal effects on SRF-, NFκB-, and AP1-mediated gene expression pathways, which are downstream of Rho family proteins [27,28,29]. In utero expression of the p.R66W variant resulted in cortical neuron migration defects and impaired axon elongation. Together, these findings suggest that the p.R66W variant disrupts RAC3 function in a variant-specific manner, leading to a severe and distinct prenatal phenotype. 

## 2. Materials and Methods

### 2.1. Plasmids

The cDNAs for human RAC3 (NM_005052.3) and PAK1, which were generous gifts from the late Prof. Alan Hall (University College London, UK), were subcloned into the pCAG-Myc vector, courtesy of Dr. T. Kawauchi (Kyoto University, Japan). The pCAG-GFP vector was obtained from Dr. Connie Cepko (Addgene plasmid # 11150). To create the RAC3-R66W and constitutively active RAC3-Q61L (GTPase-deficient) variants, mutations were introduced as previously described [22] and cloned into both pCAG-Myc and pTriEx-4 vectors (Merck, Darmstadt, Germany). RAC-binding regions (RBRs) of human PAK1 (aa 67–150), human MLK2 (aa 401–550), and rat N-WASP (aa 191–270) were amplified from a cDNA pool of U251MG cells or rat brain and inserted into the pGS21a vector (GenScript, Piscataway, NJ, USA) [21]. DNA sequencing confirmed the integrity of all constructs.

### 2.2. Antibodies

Primary antibodies used were anti-Myc (Medical & Biological Laboratories, Nagoya, Japan; Cat# M047-3, RRID: AB_591112) and anti-GFPs from Medical & Biological Laboratories (Cat# 598, RRID: AB_591819) and Nacalai Tesque (Kyoto, Japan; Cat# 04404-84, RRID: AB_10013361). Alexa Fluor 488 (Invitrogen, Carlsbad, CA, USA) was used as the secondary antibody. Rhodamine phalloidin (Invitrogen) and 4′, 6-diamidino-2-phenylindole (DAPI) (Sigma-Aldrich, Cat# D9542) were used for staining filamentous actin and DNA, respectively. 

### 2.3. GDP/GTP Exchange and GTP Hydrolysis Assays

Recombinant His-tagged RAC3 and RAC3-R66W proteins were expressed and purified according to the manufacturer’s guidelines. GTP hydrolysis activity was measured by monitoring the GTP concentration changes using a luciferase-based GTPase assay kit (GTPase-Glo Assay Kit, Promega, Madison, WI, USA), following the provided instructions [30]. The GDP/GTP exchange reaction was evaluated by quantifying the release of methylanthraniloyl (mant)-GDP (Sigma-Aldrich, St. Louis, MO, USA) [31].

### 2.4. Cell Culture and Transfection

COS7 and primary hippocampal neurons derived from embryonic day (E) 16.5 mice were cultured as described [7]. Transient transfections were conducted using the polyethyleneimine “MAX” reagent (for COS7 cells) (Polysciences Inc., Warrington, PA, USA) or the Neon transfection system (for primary neurons) (Invitrogen, Carlsbad, CA, USA).

### 2.5. Pull-Down Assay

Glutathione S-transferase (GST)-fused RBRs of PAK1, MLK2, and N-WASP were expressed in Escherichia coli BL21 (DE3) strain and purified following the manufacturer’s instructions. COS7 cells were seeded in 35 mm culture dishes and transfected with pCAG-Myc-RAC3 or -Myc-RAC3-R66W (0.3 μg each). After 24 h, cells were lysed using a pull-down buffer (50 mM Tris-HCl, pH7.5, 150 mM NaCl, 5 mM MgCl_2_, 0.1% SDS, 1% Nonidet P-40, and 0.5% deoxycholate). Following centrifugation at 15,000× *g* for 10 min at 4 °C to remove insoluble components, the supernatant was incubated with Glutathione Sepharose 4B beads (GE Healthcare Life Sciences) pre-bound with GST-PBD-PAK1, -MLK2, or N-WASP for 30 min at 4 °C. The proteins bound to the beads were then subjected to western blot analysis using an LAS-4000 luminescent image analyzer (GE Healthcare Life Sciences, Buckinghamshire, England).

### 2.6. Assay of SRF-, AP1- and NFκB-Mediated Gene Transcription [22]

COS7 cells, seeded in 24-well plates, were co-transfected with the control reporter vector and the appropriate RAC3 expression plasmid (0.05 μg per well), along with the SRF-, AP1-, or NFκB-luciferase reporter plasmid (0.2 μg/well). Following transfection, cells were rinsed with phosphate-buffered saline and lysed using passive lysis buffer in accordance with the manufacturer’s protocol. Luciferase activity was measured using the dual-luciferase reporter assay system (Promega).

### 2.7. In Utero Electroporation [21]

Pregnant ICR mice (Japan SLC, Shizuoka, Japan) were administered anesthesia using a combination of midazolam (4 mg/kg), medetomidine (0.75 mg/kg), and butorphanol (5 mg/kg) [32]. A glass micropipette was used to inject 1 µL of plasmid solution into the lateral ventricle of embryos at embryonic day 14 (E14). The embryos, still in utero, were placed between tweezer-type disc electrodes (5 mm diameter; CUY650-5, NEPA Gene, Chiba, Japan), and five electric pulses (35 V, 50 ms) were applied at 450 ms intervals using a NEPA21 electroporator (NEPA Gene) to efficiently target the somatosensory region of the parietal lobe. Brain tissues were collected postnatally at designated time points, fixed, sectioned, and prepared for further analysis. To quantify GFP-positive cell distribution, coronal sections of the labeled cortex were divided into three bins, with labeled cells counted across a minimum of three sections per brain. For axonal analysis, a plasmid solution containing 0.1 µg of pCAG-Myc-empty (control), -Myc-RAC3, or -Myc-RAC3-R66W was co-injected with 0.4 µg of pCAG-GFP. Tissues were collected at P0 and P7 for further analysis. All animal experiments were conducted during the day, with no animals excluded or harmed during the procedures. 

### 2.8. Immunofluorescence

Immunofluorescence analysis was performed as previously described [22]. Cultured cell images were captured using either a BZ-9000 microscope (Keyence, Osaka, Japan) or an LSM-880 confocal laser microscope (Carl Zeiss, Oberkochen, Germany). For tissue staining, brains were embedded in 3% agarose and sectioned into 100 μm-thick slices using a vibratome. Images were then acquired and processed. For the analyses of axon elongation in hippocampal neurons and cortical neuron migration (Figure 4A,C), 15–20 optical Z-stack confocal images (0.42 μm intervals) and 3–4 images (27 μm intervals) were obtained, respectively. Cellular morphology and fluorescence intensity levels were quantified using ImageJ software (version 2.14.0/1.54f) to assess relevant parameters.

### 2.9. Quantitative Analysis of Axonal Growth

For assessing axonal growth and extension, the GFP signal intensity of callosal axons was measured using ImageJ at P0 and P7 in distinct regions (bin 1–4) of coronal sections. The relative intensities of bins were normalized with bin 1 as 1.0, and comparisons were made using the R programming language and software environment (https://intro2r.com/citing-r.html; https://cran.r-project.org/doc/FAQ/R-FAQ.html#Citing-R; access date, 14 November 2023).

### 2.10. Statistical Analysis

Cell selection and trace assessment for all cell imaging studies were performed in a blinded manner utilizing ImageJ software. Dunnett’s or Tukey’s test was used to determine statistical significance for multiple group comparisons, while Welch’s *t*-test was used for two-group comparisons. A *p*-value of less than 0.033 was considered statistically significant. Statistical analysis was carried out using Prism 9 (GraphPad Software, Boston, MA, USA).

## 3. Results

### 3.1. Biochemical Properties of RAC3-R66W

To explore the pathophysiological features of RAC3-R66W, we examined its activation state by analyzing its intrinsic GDP/GTP exchange and GTP hydrolysis activities using recombinant proteins. Our analysis showed that the GDP/GTP exchange activity of RAC3-R66W was slightly enhanced compared to the wild-type protein, as determined by relative fluorescence readings (Figure 1A,B). In contrast, the GTP hydrolysis activity of RAC3-R66W was reduced compared to the wild type (Figure 1C,D). These observations suggest that RAC3-R66W behaves as an activated form of the protein. 

### 3.2. Biological Properties of RAC3-R66W

We examined the effects of the p.R66W variant on the morphology of primary cultured hippocampal neurons, comparing it to wild-type RAC3 and the constitutively active RAC3-Q61L variant [33,34]. Neurons expressing wild-type RAC3 showed typical differentiation patterns, extending a single axon (Figure 2A–C), suggesting that the basal activity of exogenous RAC3 does not significantly affect neuronal differentiation in vitro. On the contrary, neurons expressing RAC3-Q61L showed phenotypes consistent with activated signaling, including rounded cell morphology and lamellipodia formation, leading to increased cell solidity and impaired differentiation (Figure 2A–C). Remarkably, neurons expressing RAC3-R66W failed to extend axons and displayed similar signs of cell rounding and lamellipodia formation (Figure 2A–C). Thus, the activated state of RAC3-R66W appears to profoundly impact neuronal morphology.

### 3.3. Interaction of RAC3-R66W with Downstream Effectors PAK1, MLK2, and N-WASP

Biochemical and cell biological analyses indicated that RAC3-R66W functions as an activated variant in vitro (Figure 1 and Figure 2). To explore the underlying molecular mechanisms, we examined its effects on downstream signaling pathways by pull-down assays. We assessed the interaction between RAC3-R66W, expressed in COS7 cells, and the recombinant PBD of downstream effectors, including PAK1 (a key protein kinase), MLK2 (a MAPK kinase kinase), and N-WASP (an actin cytoskeleton regulator). The results showed that RAC3-R66W exhibited similar affinity for MLK2 as RAC3-Q61L, while its interaction with PAK1 and N-WASP was moderate (Figure 3A–D). This suggests that RAC3-R66W has a stronger interaction with MLK2 compared to PAK1 and N-WASP.

Next, we investigated gene expression pathways associated with SRF, NFκB, and AP1 using a transient expression method in COS7 cells, as Rho family members are involved in these pathways [27,28,29]. Our findings revealed that RAC3-R66W little affected the transcriptional activity of these genes compared to the wild type (Figure 3E–G). This suggests that the partial activation of PAK1 by RAC3-R66W is insufficient to fully activate downstream pathways, leading to reduced activation of SRF, NFκB, and AP1 signaling. Additional downstream effectors not assessed in this study may also play a role in RAC3-R66W signaling.

### 3.4. Effects of the p.R66W Variant on Neuronal Migration During Corticogenesis In Vivo

Based on our in vitro findings, the p.R66W variant likely promotes an activated state, leading to abnormal neuronal morphology. Given the neuronal heterotopia observed in the patient’s white matter during microscopic examination, which is associated with disrupted cell morphology and migration, we investigated the effects of this variant on neuronal migration in vivo. Using in utero electroporation, we co-electroporated pCAG-Myc (control), pCAG-Myc-RAC3, or pCAG-Myc-RAC3-R66W with pCAG-GFP into ventricular zone (VZ) progenitor cells in E14 embryonic brains (Figure 4A) and analyzed the localization of transfected cells at P0. Neurons expressing the control vector or pCAG-Myc-RAC3 migrated normally to the superficial layer (bin 1; layers II/III) of the cortical plate (CP) (Figure 4B,C). In contrast, neurons expressing pCAG-Myc-RAC3-R66W significantly remained in the VZ/subventricular zones (VZ/SVZ) and the intermediate zone (IZ) (bin 2 and 3) (Figure 4B,C). This suggests that the basal activity of RAC3 had no effects on neuronal cell migration and that normal migration requires the physiological balance between GTP- and GDP-bound states of RAC3, and that overexpression of the activated variant disrupts this balance. When the long-term effect of the p.R66W variant was examined, abnormal neuron positioning was still observed at P7 (Figure 4D,E). Notably, the transfection efficiency of each cell varied depending on the surface area exposed to the ventricular lumen (cerebrospinal fluid), where the plasmids were injected. Consequently, neurons incorporating a higher amount of plasmid experience stronger effects of the variant compared to those incorporating a lower amount. Specifically, neurons in bin 3 showed the highest GFP expression, followed by those in bin 2 and bin 1, in descending order (Figure 4F). These findings collectively suggest that the p.R66W variant likely disrupts cortical neuron migration rather than merely delaying it, highlighting the crucial role of RAC3 in this process.

**Figure 4 cells-13-02032-f004:**
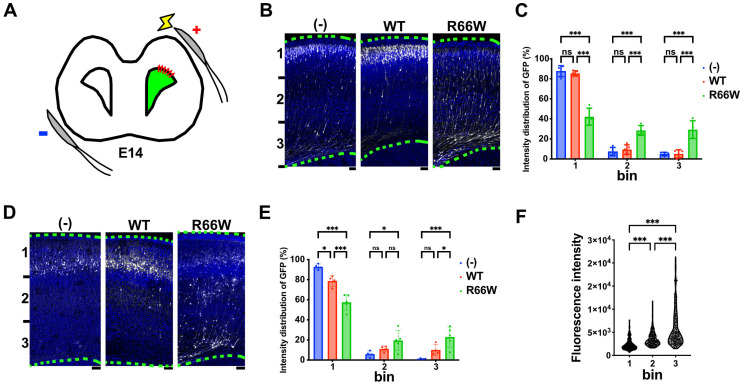
Effects of the p.R66W variant on excitatory neuron migration during corticogenesis. (**A**) Diagram illustrating in utero electroporation performed at E14. (**B**,**D**) Migration defects of neurons expressing the p.R66W variant. pCAG-GFP (0.5 μg) was co-electroporated in utero with pCAG-Myc (-), pCAG-Myc-RAC3 (WT), or -RAC3-R66W (0.1 μg each) into the VZ progenitor cells at E14.5. Coronal sections were prepared at P0 (**B**) or P7 (**D**). Coronal slices were double-stained with anti-GFP (white) and DAPI (blue). Scale bars, (**B**) 50 μm, (**D**) 100 μm. (**C**,**E**) Quantification of the distribution of GFP-positive neurons in distinct regions of the cerebral cortex (bin 1–3) at P0 (**C**) and P7 (**E**). Number of replicates, N ≥ 4. Statistical significance between WT and each variant was determined using two-way ANOVA followed by Dunnett’s post hoc test and shown with interleaved scatter with bars. (**C**) (bin 1) (-) vs. WT, *p* = 0.82; (-) vs. R66W, *p* < 0.001; WT vs. R66W, *p* < 0.001. (bin 2) (-) vs. WT, *p* = 0.84; (-) vs. R66W, *p* < 0.001; WT vs. R66W, *p* < 0.001. (bin 3) (-) vs. WT, *p* = 0.99; (-) vs. R66W, *p* < 0.001; WT vs. R66W, *p* < 0.001. (**E**) (bin 1) (-) vs. WT, *p* = 0.01; (-) vs. R66W, *p* < 0.001; WT vs. R66W, *p* < 0.001. (bin 2) (-) vs. WT, *p* = 0.51; (-) vs. R66W, *p* = 0.01; WT vs. R66W, *p* = 0.16. (bin 3) (-) vs. WT, *p* = 0.15; (-) vs. R66W, *p* < 0.001; WT vs. R66W, *p* < 0.02. (**F**) GFP intensity in individual cells expressing Rac3-R66W in bin 1–3 at P0. Number, N ≥ 149. Statistical significance was determined using one-way ANOVA with Tukey’s post hoc test (*p* < 0.033). bin 1 vs. bin 2, *p* < 0.001; bin 2 vs. bin 3, *p* < 0.001; bin 2 vs. bin 3, *p* < 0.001. * *p* < 0.033, *** *p* < 0.001. ns, not significant.

### 3.5. Effects of the p.R66W Variant on Axon Growth During Cortical Development In Vivo

The corpus callosum agenesis observed in the prenatal case harboring the R66W change suggests that this variant impairs axonal growth in vivo. To examine this, we assessed the impact of the variant on axonal growth in pyramidal neurons of layers II/III within the parietal lobe during cortical development. Using in utero electroporation at E14, we introduced pCAG-GFP along with pCAG-Myc-RAC3 (control) or pCAG-Myc-RAC3-R66W. Visualization of the corpus callosum at P0 showed significantly delayed axonal extension in RAC3-R66W-expressing neurons compared to controls (Figure 5A,B). Specifically, these neurons failed to project axons compared to control cells, which start to extend axon bundles (Figure 5A,B). However, examining the long-term effects at P7, we observed that not only control neurons but also RAC3-R66W-expressing neurons extended their axon bundles into the contralateral white matter (Figure 5C,D). These results suggest that the RAC3-R66W variant delays axonal elongation in cortical neurons but does not prevent it.

## 4. Discussion

Deleterious de novo variants in the *RAC3* gene have been linked to a complex neurodevelopmental syndrome known as NEDBAF, which is characterized by structural brain abnormalities and dysmorphic facial features. Recently, the new *RAC3* variant c.196C>T, p.R66W was identified in a male fetus at 26 weeks’ gestation, displaying both cortical and extracortical manifestations, including fetal akinesia deformation sequence and severe brain malformations such as agenesis of the corpus callosum, diencephalosynapsis, a kinked brainstem, and vermian hypoplasia [26]. This variant is situated in the Switch II region, a critical domain for interactions with effector and regulatory proteins in small GTPases. Given the distinct clinical features of this case compared to those seen in NEDBAF patients, we sought to investigate how the p.R66W variant impacts RAC3 function and contributes to this fetopathy.

Our biochemical studies demonstrated that the p.R66W variant reduces GTPase activity while slightly enhancing the intrinsic GDP/GTP-exchange activity, indicating that RAC3-R66W behaves as an activated form of the protein (Figure 1). Active variants of small GTPases are generally classified into two types: GTPase-deficient variants and fast-cycling variants, the latter characterized by an increased intrinsic GDP/GTP exchange rate. Based on our findings, we propose that the p.R66W variant falls into the GTPase-deficient category. We assume that the activation of this variant is weaker than that of the well-characterized constitutively active RAC3-Q61L variant, as the effect of RAC3-R66W on hippocampal neuron differentiation was more moderate compared to RAC3-Q61L (Figure 2).

Further analysis revealed that RAC3-R66W interacts with downstream effectors PAK1, MLK2, and N-WASP, consistent with its activated state. However, the affinities to PAK1 and N-WASP were weaker compared to RAC3-Q61L (Figure 3A–D). These observations suggest that RAC3-R66W engages with the PAK1-, MLK2-, and N-WASP-mediated signaling pathways. Notably, however, RAC3-R66W did not enhance SRF-, NFκB-, or AP1-mediated gene expression (Figure 3E–G). We speculate that this deficiency in activating SRF, NFκB, and AP1 may impair neurogenesis and contribute to the arhinencephaly observed in the patient [35,36,37,38]. Furthermore, the weaker interactions of RAC3-R66W with PAK1 and N-WASP may underlie the neurodevelopmental abnormalities observed in this case, as these effectors are involved in critical neuronal processes such as axonal growth, neurite extension, and dendrite formation [39,40]. In cortical neurons, acute expression of RAC3-R66W caused migration defects (Figure 4) and delays in the extension of callosal axons to the contralateral hemisphere (Figure 5). Additionally, the activation of alternative effector pathways by RAC3-R66W may contribute to the observed corticogenesis defects. 

RAC3-Q61L, like RAC3-R66W, disrupts neuronal migration during corticogenesis but appears to activate distinct downstream effectors (Figure 3A). Notably, the migration defects caused by RAC3-Q61L were not alleviated by dominant-negative (DN) PAK1 or DN-MLK2, indicating that these kinases are unlikely to contribute to the observed phenotype [21]. In contrast, DN-PAK1 rescued the migration defects associated with the p.E62del, p.D63N, and p.Y64C variants, which are also situated in the Switch II region alongside p.Q61L and p.R66W [21]. This strongly implicates PAK1 involvement in the defects caused by the three variants. These observations underscore the complexity of RAC3 signaling pathways and suggest that similar migration phenotypes may result from distinct molecular mechanisms. Further investigations are essential to clarify how these variants differentially disrupt neuronal migration at the molecular level.

The p.Q61L, p.E62del, p.D63N, and p.Y64C variants also impaired axon bundle elongation at P0 [21]. Interestingly, at P7, axon growth was absent in neurons expressing RAC3-Q61L, -E62del, and -Y64C, whereas neurons expressing RAC3-D63N extended axons to the contralateral cortex, similar to those expressing RAC3-R66W. Notably, the defective phenotypes caused by the p.E62del, p.D63N, and p.Y64C variants, but not by p.Q61L, were rescued by DN-PAK1, reflecting the rescue observed in cortical neuron migration [21]. Although the downstream signaling mechanisms underlying the p.R66W-dependent phenotype remain unclear, these findings further emphasize that individual variants disrupt downstream signaling in a variant-specific manner, yet can lead to similar brain morphological abnormalities.

While the activation of PAK1, MLK2, and N-WASP by RAC3-R66W likely plays an essential role in the pathophysiology of the severe prenatal manifestations, these effectors do not seem to influence SRF, NFκB, or AP1-mediated gene expression, which are essential for cell growth and proliferation. Moreover, RAC3 interacts with numerous effectors beyond PAK1, MLK2, and N-WASP, suggesting that the p.R66W variant may disrupt other downstream signaling pathways. To gain more comprehensive understanding of the pathogenicity of the p.R66W variant, further studies are needed to map its complete interactome and assess its effects on different effector systems. Although *RAC3* expression is highly enriched in brain tissues, the extracerebral features observed in the studied prenatal case (e.g., clenched hands, overlapping fingers, and congenital talipes equinovarus) suggest that the p.R66W variant may also impact the development and function of other organs as well [26]. In this context, the *RAC1* p.Y40H variant has been linked to VACTERL association, a disorder defined by anomalies in the vertebrae, anus, heart, kidneys, trachea, esophagus, and limbs [41]. These observations expand the phenotypic spectrum of RAC-related disorders to include more complex conditions featuring extracerebral manifestations, likely underpinned by specific molecular mechanisms. Although preliminary, these findings support the importance of RAC-related disorders in the differential diagnosis of complex malformative cases in prenatal setting. The identification and characterization of additional non-canonical *RAC3* variants will be crucial to define the full spectrum of these atypical phenotypes and provide insights into the tissue-specific functions of RAC3.

In summary, our research broadens the current knowledge of the genetic and phenotypic spectrum of RAC3-related disorders and sheds light on the molecular mechanisms driving this condition. We propose that the p.R66W variant acts as an active form of RAC3, potentially activating the PAK1, MLK2, and N-WASP signaling pathways. In addition, this variant failed to activate gene expression by SRF, NFκB, and AP1. While the primary mechanisms underlying the patient’s neurological and extracerebral manifestations remain to be fully elucidated, the molecular disruptions resulting from the p.R66W variant significantly contribute to the pathogenesis of this complex prenatal phenotype. It is also possible that RAC3-R66W activates other downstream targets that were not investigated in this study. Additionally, our findings underscore the importance of the Switch II region in modulating RAC3 activity during brain development. Future studies aiming at further mapping the RAC3 signaling network and understanding how pathogenic variants perturb these pathways may shed light on the pathophysiology of atypical clinical phenotypes and open new avenues for the potential development of targeted therapies.

## 5. Conclusions

This study provides new insights into the pathophysiological mechanisms underlying the *RAC3* p.R66W variant, a novel variant associated with a distinct and severe prenatal phenotype featuring cortical and extracortical anomalies. In vitro and in vivo experiments demonstrated that the p.R66W variant functions as an activated form of RAC3 and disrupts cortical development by impairing neuronal migration and axon elongation. Notably, the variant induces these phenotypes through specific molecular mechanisms distinct from other *RAC3* pathogenic variants, emphasizing the complexity and diversity of RAC3-mediated signaling pathways. The findings in this study extend the phenotypic and molecular spectrum of RAC3-related disorders, highlighting the importance of the Switch II region in regulating RAC3 activity. Future studies are needed to further map the interactome of RAC3-R66W and investigate its effects on extracerebral tissues to fully elucidate its role in this severe fetopathy. These insights may ultimately contribute to the development of targeted therapeutic strategies for RAC3-related disorders.

## Figures and Tables

**Figure 1 cells-13-02032-f001:**
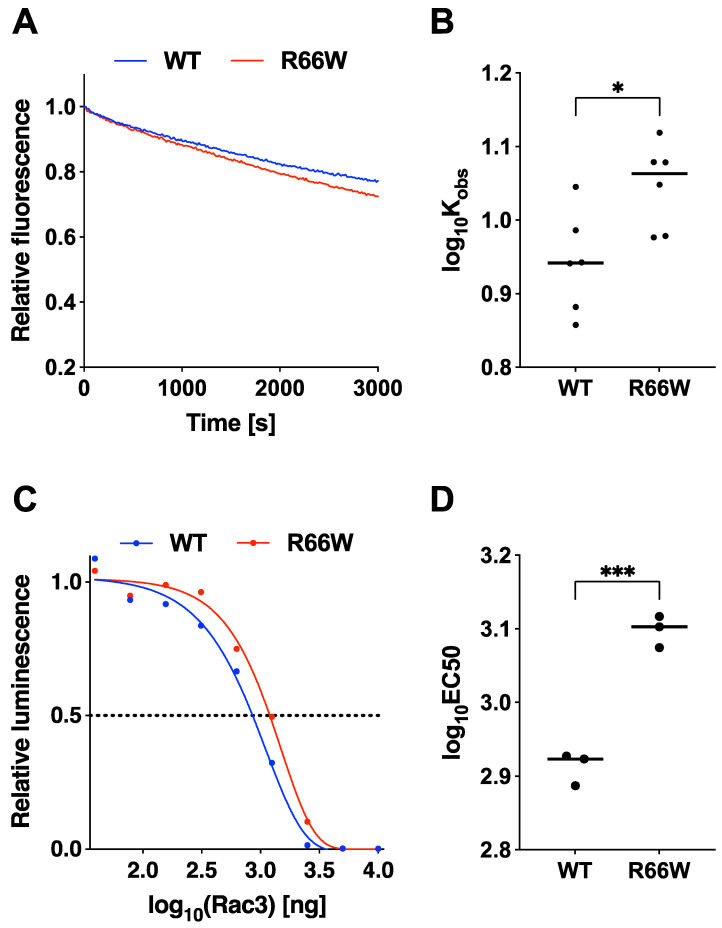
Effects of the p.R66W variant on GDP/GTP-exchange and GTP-hydrolysis activities of RAC3. (**A**) GDP/GTP-exchange activity: Recombinant His-tagged RAC3 (WT) and RAC3-R66W (R66W) proteins were preloaded with fluorescent ^mant^GDP and incubated with non-hydrolysable GTP analog, monitoring relative fluorescence over time. (**B**) ^mant^GDP-dissociation rates: Dissociation rates of WT and R66W were determined as observed rate constants (Kobs [×10^−5^ s^−1^]) based on the data in (**A**). Samples sizes: WT, N = 6; R66W, N = 6. Different letters above boxes denote statistically significant differences (*p* < 0.033) by Tukey’s test. R66W vs. WT, *p* = 0.02. * *p* < 0.05. (**C**) GTP-hydrolysis activity: GTPase activity of His-RAC3 (WT) and -RAC3-R66W (R66W) was measured by tracking GTP concentration changes using a GTPase-Glo assay kit. (**D**) EC50 values: Half maximal effective concentration (EC50) values were derived from the sigmoidal fitting curve in (**C**). Sample sizes: WT, N = 3; R66W, N = 3. Statistical significance was calculated as in (**B**), with R66W vs. WT at *p* = 0.0005. *** *p* < 0.001.

**Figure 2 cells-13-02032-f002:**
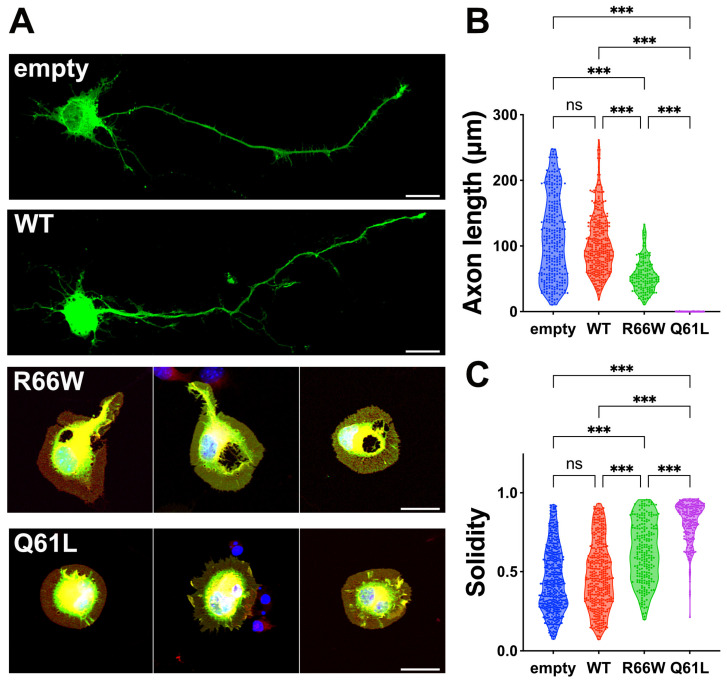
Effects of the p.R66W variant on neuron morphology in vitro. (**A**) Primary hippocampal neurons harvested from E16 embryos were co-electroporated with pCAG-GFP (0.1 μg) along with pCAG-Myc (-), pCAG-Myc-RAC3 (WT), RAC3-R66W, or RAC3-Q61L (0.3 μg each). After 3 days in vitro, the cells were fixed and co-stained with anti-GFP (green), rhodamine phalloidin (red) and DAPI (blue). Scale bars, 10 μm. (**B**,**C**) Quantification of neuron morphology from (**A**). (**B**) Length of axon (the longest neurite) of GFP-positive neurons shown as violin plots with dots. The number of neurons was as follows: (–), N = 280; WT, N = 324; R66W, N = 145; Q61L, N = 133. Statistical significance was determined using one-way ANOVA with Tukey’s post hoc test (*p* < 0.033). (-) vs. WT, *p* = 0.3; (-) vs. R66W, *p* < 0.001; (-) vs. Q61L, *p* < 0.001; WT vs. R66W, *p* < 0.001; WT vs. Q61L, *p* < 0.001; R66W vs. Q61L, *p* < 0.001. (**C**) Cell solidity of GFP-positive neurons was shown in violin plots with boxplots. “Solidity” is the ratio of the area of a cell to the area of a convex hull of the cell [16]. The number of neurons was as follows: (–), N = 4825; WT, N = 392; R66W, N = 231; Q61L, N = 205. Statistical significance was determined using one-way ANOVA with Tukey’s post hoc test (*p* < 0.033). (-) vs. WT, *p* = 0.26; (-) vs. R66W, *p* < 0.001; (-) vs. Q61L, *p* < 0.001; WT vs. R66W, *p* < 0.001; WT vs. Q61L, *p* < 0.001; R66W vs. Q61L, *p* < 0.001. *** *p* < 0.001. ns, not significant.

**Figure 3 cells-13-02032-f003:**
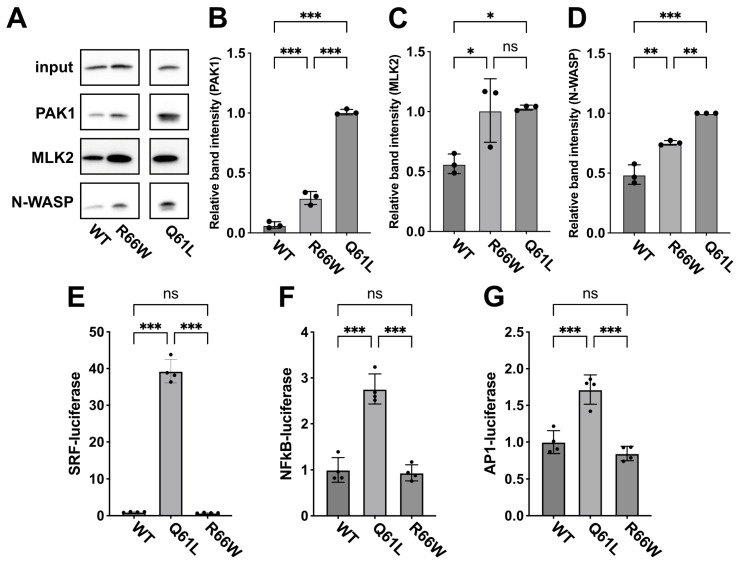
Effects of the p.R66W variant on RAC3 interactions with downstream signaling pathways. (**A**) Evaluation of binding to the PBD of PAK1, MLK2, and N-WASP. COS7 cells were transfected with pCAG-Myc-RAC3 (WT) or -RAC3-R66W (R66W) (0.3 μg each). Pull-down assays utilizing GST-PAK1-PBD, -MLK2, or -N-WASP (5 μg each) were performed as detailed in the “Materials and methods”. Bound RAC3 was detected by western blotting with anti-Myc, and total cell lysates were also probed with anti-Myc to ensure proper normalization (Input). Uncropped blotting data are shown in Appendix A. (**B**–**D**) Quantification of RAC3 bound to GST-PBD-PAK1 (**B**), GST-PBD-MLK2 (**C**), or GST-PBD-N-WASP (**D**) was carried out. The relative intensity of the bands is displayed, with RAC3-Q61L set to a reference value of 1.0. N = 3 replicates. Statistical significance was determined using one-way ANOVA with Tukey’s post hoc test (*p* < 0.033). (**B**) WT vs. R66W, *p* < 0.001; WT vs. Q61L, *p* < 0.001; R66W vs. Q61L, *p* < 0.001. (**C**) WT vs. R66W, *p* = 0.03; WT vs. Q61L, *p* = 0.03; R66W vs. Q61L, *p* = 0.98. (**D**) WT vs. R66W, *p* = 0.001; WT vs. Q61L, *p* < 0.001; R66W vs. Q61L, *p* < 0.002. * *p* < 0.033, ** *p* < 0.002, *** *p* < 0.001. (**E**–**G**) Effects of the p.R66W variant on SRF-, NFkB- and AP1-dependent gene transcription. COS7 cells were co-transfected with pCAG-Myc, -Myc-RAC3-WT, and -RAC3-R66W (0.1 μg each/well) in various combinations, along with luciferase reporter plasmids for SRF, NFkB, or AP1 (0.05 μg each/well). The Luciferase activity from the wild-type control was defined as 1.0, with relative activities presented as scatter plots with bars. N = 4 replicates. Statistical significance was determined using one-way ANOVA with Tukey’s post hoc test (*p* < 0.033). (**E**) WT vs. Q61L, *p* < 0.001; WT vs. R66W, *p* = 0.98; Q61L vs. R66W, *p* < 0.001. (**F**) WT vs. Q61L, *p* < 0.001; WT vs. R66W, *p* = 0.94; Q61L vs. R66W, *p* < 0.001. (**G**) WT vs. Q61L, *p* < 0.001; WT vs. R66W, *p* = 0.39; Q61L vs. R66W, *p* < 0.001. ns, not significant.

**Figure 5 cells-13-02032-f005:**
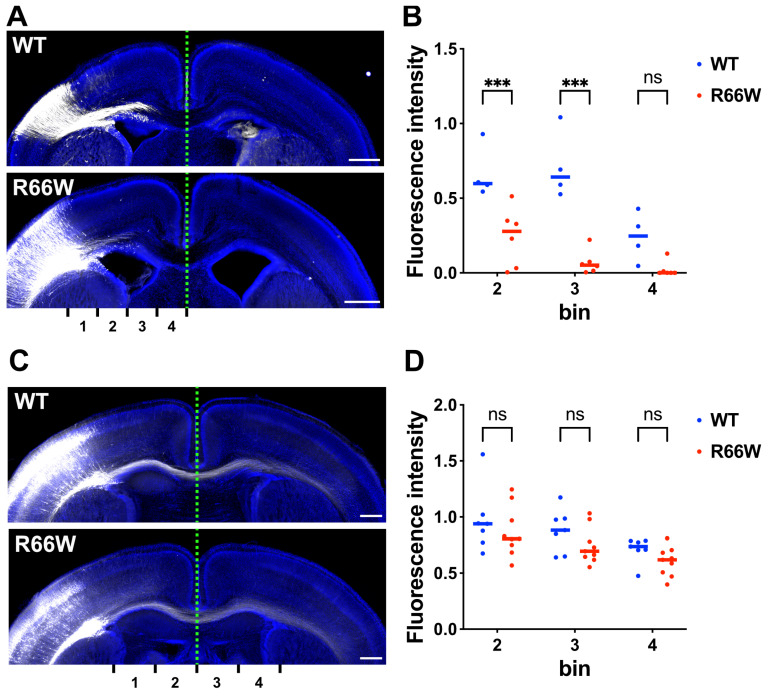
Effects of the p.R66W variant on axonal extension during cortical development in vivo. (**A**,**C**) pCAG-GFP was co-electroporated at E14 with either pCAG-Myc-RAC3 (WT) or -RAC3-R66W (0.1 μg each). Coronal sections were prepared at P0 (**A**) or P7 (**C**) and visualized using GFP (white). DAPI staining (blue) of a slice is also shown. Scale bars, 500 μm (**A**,**C**). (**B**,**D**) The GFP intensity of the callosal axon was measured at P0 (**B**) or P7 (**D**) in different regions (bin 1–4), and then the relative intensities of bins were normalized with bin 1 as 1.0. Statistical significance between WT and each variant was determined using two-way ANOVA and shown with interleaved scatter with bars. (**B**) The number of brains was as follows: WT, N = 4; R66W, N = 6. bin 2, *p* < 0.001; bin 3, *p* < 0.001; bin 4, *p* = 0.11. (**D**) The number of brains was as follows: WT, N = 7; R66W, N = 9. bin 2, *p* = 0.66; bin 3, *p* = 0.42; bin 4, *p* = 0.59. *** *p* < 0.001. ns, not significant.

## Data Availability

The data that support the findings of this study are available from the corresponding authors, upon reasonable request.

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
