# Peer review of "The p.R66W Variant in RAC3 Causes Severe Fetopathy Through Variant-Specific Mechanisms"

_cells, 2024, doi:10.3390/cells13232032_

Round 1

Reviewer 1 Report

Comments and Suggestions for Authors

This manuscript by Sugawara et al describes novel pathophysiological functions of RAC3 R66W variant which recently identified the mutation in a male fetus, exhibited complex brain malformations etc. Using biochemical approaches, the authors characterized the biochemical RAC3 R66W activities and found R66W variant acts as an active form of RAC3. Furthermore, the authors found that the acute expression of the mutant caused migration defects in cortical neurons and delayed the extension of callosal axon to the contralateral hemisphere, indicating the impact of RAC3 R66W in neuronal development and fetopathy.

Specific comments:

1. page 4 (Line 176) “These findings strongly suggest that RAC3-R66W behaves as a constitutively active variant.”

Perhaps it could be better to slightly tone down the sentence (RAC3-R66W behaves as a constitutively active variant) following their discussion (RAC3-R66W behaves as an activated form of the protein). Compared to RAC3 Q61L which the authors used as a constitutive active form in Figure 2-3, the R66W GTP hydrolysis activity has not much eliminated as they reported.

2. Fig 2A legend, “At 3 days in vitro, the cells were fixed and immunostained for GFP (Green).”,

The R66W and Q61L images contain red and blue colors in addition to the green color. The authors should update the legend. Perhaps “co-stained with anti-GFP (green), rhodaminephalloidin (red) and DAPI (blue) according to the authors’ previous report (https://doi.org/10.1093/brain/awac106)?

3. Also, in Fig 2A legend, Scale bars, 10 um.”

The size of scale bar (10 um) on the images (cell body size is ~5 um) is quite different from the authors' previous reported similar images (cell body size is ~20 um) (https://doi.org/10.1093/brain/awac106), please confirm the scale bar is correct.

4. In Figure 2 B, C, Figure 3B-G, Figure 4B, D in the PDF manuscript file, I couldn’t read the corrupted symbols for statistical significance. Probably “***” ?

5. Fig 2C legend, “Cell 215 solidity of GFP-positive neurons shown as violin plots with boxplots.”

How to define and calculate the solidity? The authors should describe it in the figure legend or in the method section and cite their previous paper.

6. Page 8 (Line 266-268) “Using in utero electroporation, we co-electroporated pCAG-Myc (control), pCAG-Myc-RAC3, or pCAG-Myc-RAC3-R66W with pCAG-EGFP into ventricular zone (VZ) progenitor cells in E14.5 embryonic brains and analyzed the localization of transfected cells at P0.”

Does the pCAG promoter with the electroporation to the progenitor cells can specifically express these Myc fusion proteins with EGFP into the neurons? No differentiation into other cells such as glial cells (through radial glial progenitor cells etc)?

7. Page 8 (Line 276-279) “Neurons with high expression levels of the variant tended to accumulate in bin 3. In contrast, transfection efficiency for each cell likely varied based on the cell surface area exposed to the ventricular lumen (cerebrospinal fluid) where plasmids were injected, meaning that cells with lower levels of the variant may have experienced only partial effects.”

It would be helpful if the authors prepared an additional bar graph to directly compare the GFP fluorescent intensity of individual neurons between bin 1-3.

8. Related, the coexpression levels between EGFP and RAC3 variants are practically proportional using in utero electroporation?

9. Also, the authors should discuss the difference of the migration results between RAC-R66W and RAC3-Q61L (the authors have previously reported).

10. Similarly, I would encourage the authors to expand their discussion/speculation about the potential mechanisms of the R66W-dependent migration defect and delay in the extension along with the other RAC3 variants including the Q61L.

Author Response

Specific comments:

  1. page 4 (Line 176) “These findings strongly suggest that RAC3-R66W behaves as a constitutively active variant.”

Perhaps it could be better to slightly tone down the sentence (RAC3-R66W behaves as a constitutively active variant) following their discussion (RAC3-R66W behaves as an activated form of the protein). Compared to RAC3 Q61L which the authors used as a constitutive active form in Figure 2-3, the R66W GTP hydrolysis activity has not much eliminated as they reported.

>  We agree to the reviewer’s comment. We toned down the down the expression in the new version (p. 4, Line 182).

  1. Fig 2A legend, “At 3 days in vitro, the cells were fixed and immunostained for GFP (Green).”,

The R66W and Q61L images contain red and blue colors in addition to the green color. The authors should update the legend. Perhaps “co-stained with anti-GFP (green), rhodamine phalloidin (red) and DAPI (blue) according to the authors’ previous report (https://doi.org/10.1093/brain/awac106)?

> Please accept our apologies for this basic error. As the reviewer correctly noted, the text has been corrected (p. 6, Line 211 – 212). We also added the info of rhodamine phalloidin (red) and DAPI (blue) in the Materials section of the new manuscript (p. 3, Line 101-103).

  1. Also, in Fig 2A legend, “Scale bars, 10 um.”

The size of scale bar (10 um) on the images (cell body size is ~5 um) is quite different from the authors' previous reported similar images (cell body size is ~20 um) (https://doi.org/10.1093/brain/awac106), please confirm the scale bar is correct.

> Following the reviewer's comment, we have checked the figure and found the scale to be incorrect. We apologize for this mistake and have amended it to the correct scale (Fig 2A in the revised manuscript).

  1. In Figure 2 B, C, Figure 3B-G, Figure 4B, D in the PDF manuscript file, I couldn’t read the corrupted symbols for statistical significance. Probably “***” ?

> This concern was also raised by another reviewer. The garbled symbols indeed represent “***”. While the statistical significance “***” was correctly displayed in the Word and PDF files we originally submitted, it appears to have been corrupted in the PDF file generated through the journal's website. We have replaced the problematic figures in the revised manuscript. If the issue persists, we will consult with the editorial office to find a solution. We hope that the corrected version is now available.

  1. Fig 2C legend, “Cell solidity of GFP-positive neurons shown as violin plots with boxplots.”

How to define and calculate the solidity? The authors should describe it in the figure legend or in the method section and cite their previous paper.

>  As suggested by the reviewer, we have described how the solidity was defined and calculated in the Method section of the new manuscript (p.7, line 218 – 219). We have also cited the previous paper as [16].

  1. Page 8 (Line 266-268) “Using in utero electroporation, we co-electroporated pCAG-Myc (control), pCAG-Myc-RAC3, or pCAG-Myc-RAC3-R66W with pCAG-EGFP into ventricular zone (VZ) progenitor cells in E14.5 embryonic brains and analyzed the localization of transfected cells at P0.”

Does the pCAG promoter with the electroporation to the progenitor cells can specifically express these Myc fusion proteins with EGFP into the neurons? No differentiation into other cells such as glial cells (through radial glial progenitor cells etc)?

> At the time of electroporation at E14, cells generated from ventricular zone (VZ) progenitor/stem cells are committed exclusively to becoming cortical neurons. Astrocytes, on the other hand, begin to be generated after E15. In addition, the CAG promoter functions predominantly in post-mitotic cells, such as neurons, but not in radial glia or astrocytes. It is noteworthy that cells committed to becoming neurons are post-mitotic and do not re-enter the cell cycle, whereas cells committed to becoming astrocytes can continue dividing as they migrate to their final positions. Taken together, it is widely accepted that these Myc fusion proteins and GFP are specifically expressed in neurons rather than in glial cells under these conditions. We hope the reviewer will kindly consider and accept this explanation.

  1. Page 8 (Line 276-279) “Neurons with high expression levels of the variant tended to accumulate in bin 3. In contrast, transfection efficiency for each cell likely varied based on the cell surface area exposed to the ventricular lumen (cerebrospinal fluid) where plasmids were injected, meaning that cells with lower levels of the variant may have experienced only partial effects.”

It would be helpful if the authors prepared an additional bar graph to directly compare the GFP fluorescent intensity of individual neurons between bin 1-3.

> In response to the reviewer’s suggestion, we have directly compared the GFP fluorescent intensity of individual neurons between bins 1–3 of RAC3-R66W-expressing cells. This analysis has been included in the revised manuscript as Fig. 4F. Correspondingly, the results section (p. 9, line 289 – p.10, line 305) and figure legend has been updated in the revised manuscript.

  1. Related, the coexpression levels between EGFP and RAC3 variants are practically proportional using in utero electroporation?

> We assume that the expression levels of EGFP are practically proportional to those of the co-expressed RAC3 variant in the in utero electroporation experiments. This assumption is based on the following reasons: (1) all proteins were expressed using the same vector driven by the CAG promoter, and (2) the transfection efficiency for each plasmid is consistent, as the cell surface area exposed to the ventricular lumen—where the plasmids are injected—is the same for all co-injected plasmids. We hope the reviewer will kindly consider and accept this explanation.

  1. Also, the authors should discuss the difference of the migration results between RAC-R66W and RAC3-Q61L (the authors have previously reported).

> According to the reviewer’s suggestion, we included a discussion on the differences in migration results between RAC3-R66W and RAC3-Q61L, and other 3 variants in the revised manuscript (p.12, line 366 – 376).

  1. Similarly, I would encourage the authors to expand their discussion/speculation about the potential mechanisms of the R66W-dependent migration defect and delay in the axon extension along with the other RAC3 variants including the Q61L.

> Based on the reviewer’s suggestion, we included a discussion on the potential mechanisms of the R66W-dependent delay in the axon extension along with the other 4 variants including the Q61L in the revised manuscript (p.12, line 377 – 386).

Reviewer 2 Report

Comments and Suggestions for Authors

In this study, Sugawara and colleagues, investigated the a de novo mutation of P.R66W in the brain development. They characterized the GTPase activity and several protein interaction partners of the RAC3-R66W. Furthermore, they used in utero electroporation technique and found expression of RAC3-R66W affects neuronal migration, axonal elongation, and dendritic arborization. This study provides some etiological evidences on RAC3-R66W in related brain diseases. 

1.     The font and size of text are inconsistent in all figures. The authors should carefully review all figures and make sure they match the legend parts.

2.     The reviewer could not see the statistical marks in all figures. 

3.     Figure 3A, are the figures represent data from one blot? if not, it is not a reliable to compared findings of R66W and Q61L. Figure 3B-D, for all these 3 figures, for each bar, 9 data points are used, which is not consistent in 3C (R66W and R61L bars).

4.     Figure5C-D, it seems to the reviewer that fluorescence intensities of bins #2,3,4 are higher in R66W samples. 

Author Response

  1. The font and size of text are inconsistent in all figures. The authors should carefully review all figures and make sure they match the legend parts.

> While the font and text size were correctly displayed in the Word and PDF files originally submitted, they appear to be inconsistent in the PDF file generated on the journal's website. We have completely replaced the problematic figures in the revised manuscript. If the issue persists, we will contact the editorial office for resolution. We hope that the correct version is now accessible.

  1. The reviewer could not see the statistical marks in all figures. 

> This concern was also raised by another reviewer. The garbled symbols indeed represent “***”. While the statistical significance “***” was correctly displayed in the Word and PDF files we originally submitted, it appears to have been corrupted in the PDF file generated through the journal's website. We have replaced the problematic figures in the revised manuscript. If the issue persists, we will consult with the editorial office to find a solution. We hope that the corrected version is now available.

  1. Figure 3A, are the figures represent data from one blot? if not, it is not a reliable to compared findings of R66W and Q61L. Figure 3B-D, for all these 3 figures, for each bar, 9 data points are used, which is not consistent in 3C (R66W and R61L bars).

> As shown in Supplementary Figure S1, all the figures in Figure 3A are derived from a single blot. Regarding the data points in Figure 3B–D, the description in the previous manuscript was incorrect. Although we performed the pull-down assay 3 times, we made mistake during the quantification analysis, resulting in calculating the protein-band intensity a few times with different exposure conditions and mixed them up in the graph. We apologize this mistake and have corrected the graphs and figure legend for Figure 3B-D in the revised manuscript accordingly. We renewed Supplementary Figure S1 in the revised manuscript and added all pull-down WB data.

  1. Figure 5C-D, it seems to the reviewer that fluorescence intensities of bins #2,3,4 are higher in R66W samples.

> We agree with the reviewer’s observation. The fluorescence intensity of the R66W sample in Figure 5C appears higher, suggesting more abundant axons are labeled in this slice. However, Figure 5D shows the relative intensities of the bins in the same sample, normalized with bin 1 set to 1.0. Therefore, the comparison of intensities among the bins within each panel is critical for interpretation. We, however, agree with the concern by the reviewer. To reduce potential misunderstandings for readers, we replaced the images in Figure 5C in the revised manuscript.

Reviewer 3 Report

Comments and Suggestions for Authors

The paper entitled: "Pathophysiological significance of the p.R66W variant in RAC3 responsible for cerebral and extra-cerebral anomalies" is well written and has a good potential to be published in Cells journal. 

The authors show that the novel variant in RAC3 has a negative impact on brain development.  Interestingly, RAC3-R66W interacted with the downstream effectors PAK1, MLK2, and N-WASP.

The authors revealed that overexpression of RAC3-R66W inhibits the differentiation of primary cultured hippocampal neurons and results in impairments in cortical neuron migration and axonal elongation during cortex development. 

I have criticism for this work:

1) In the method section, the authors need to clearly indicate which sequence they used for in-utero electroporation. Is it a human variant?

2) In Fig 4, the authors need to add a diagram showing the area of electroporation at e14.5. 

3) Fig 2a, Fig4, Fig5 A,C. Are they Z-stacks?

Author Response

1) In the method section, the authors need to clearly indicate which sequence they used for in-utero electroporation. Is it a human variant?

> We use the human RAC3-R66W variant in all experiments. We put the accession number NM_005052.3, which represents the mRNA transcribed from the human RAC3 gene, in the revised manuscript (p.2, line 86). 

2) In Fig 4, the authors need to add a diagram showing the area of electroporation at e14.5.

> According to the reviewer’s suggestion, added a diagram in the revised manuscript as Fig. 4A. We also added some description in the figure legend (p.9, line 289-290).

3) Fig 2a, Fig4, Fig5 A,C. Are they Z-stacks?

> Pictures of Fig 2A and Fig 4 A,C are captured using Z-stacks. We clarified this in the revised manuscript (p.4, lines 156 – 159).

Round 2

Reviewer 2 Report

Comments and Suggestions for Authors

the revised manuscript is greatly improved. thus, I support the acceptance of current version of manuscript.

Reviewer 3 Report

Comments and Suggestions for Authors

I recommend to accept the paper in its present form